# The Role of Fatty Acids in Ceramide Pathways and Their Influence on Hypothalamic Regulation of Energy Balance: A Systematic Review

**DOI:** 10.3390/ijms22105357

**Published:** 2021-05-19

**Authors:** Andressa Reginato, Alana Carolina Costa Veras, Mayara da Nóbrega Baqueiro, Carolina Panzarin, Beatriz Piatezzi Siqueira, Marciane Milanski, Patrícia Cristina Lisboa, Adriana Souza Torsoni

**Affiliations:** 1Biology Institute, State University of Rio de Janeiro, UERJ, Rio de Janeiro 20551-030, Brazil; pclisboa.uerj@gmail.com; 2Faculty of Applied Science, University of Campinas, UNICAMP, Campinas 13484-350, Brazil; alana_veras_@hotmail.com (A.C.C.V.); mayarabaqueiro2009@hotmail.com (M.d.N.B.); carolinapanzarin@hotmail.com (C.P.); bia.piatezzi@gmail.com (B.P.S.); marciane.milanski@fca.unicamp.br (M.M.); 3Obesity and Comorbidities Research Center, University of Campinas, UNICAMP, Campinas 13083-864, Brazil

**Keywords:** sphingolipids, lipotoxicity, central nervous system

## Abstract

Obesity is a global health issue for which no major effective treatments have been well established. High-fat diet consumption is closely related to the development of obesity because it negatively modulates the hypothalamic control of food intake due to metaflammation and lipotoxicity. The use of animal models, such as rodents, in conjunction with in vitro models of hypothalamic cells, can enhance the understanding of hypothalamic functions related to the control of energy balance, thereby providing knowledge about the impact of diet on the hypothalamus, in addition to targets for the development of new drugs that can be used in humans to decrease body weight. Recently, sphingolipids were described as having a lipotoxic effect in peripheral tissues and the central nervous system. Specifically, lipid overload, mainly from long-chain saturated fatty acids, such as palmitate, leads to excessive ceramide levels that can be sensed by the hypothalamus, triggering the dysregulation of energy balance control. However, no systematic review has been undertaken regarding studies of sphingolipids, particularly ceramide and sphingosine-1-phosphate (S1P), the hypothalamus, and obesity. This review confirms that ceramides are associated with hypothalamic dysfunction in response to metaflammation, endoplasmic reticulum (ER) stress, and lipotoxicity, leading to insulin/leptin resistance. However, in contrast to ceramide, S1P appears to be a central satiety factor in the hypothalamus. Thus, our work describes current evidence related to sphingolipids and their role in hypothalamic energy balance control. Hypothetically, the manipulation of sphingolipid levels could be useful in enabling clinicians to treat obesity, particularly by decreasing ceramide levels and the inflammation/endoplasmic reticulum stress induced in response to overfeeding with saturated fatty acids.

## 1. Background

Obesity has become a major public health issue, and an effective therapy has yet to be well established. Individuals who gain excess weight are at risk of developing chronic diseases, such as type 2 diabetes mellitus, cardiovascular diseases, and cancer [1,2]. Recently, obese individuals were described as being more predisposed to developing severe symptoms of SARS-CoV-2, reinforcing to the scientific community that there is a need to improve the understanding of the pathophysiology of obesity and develop novel treatment strategies [3].

Excessive weight gain occurs when caloric consumption exceeds energy expenditure. Energy homeostasis is tightly regulated by the hypothalamus, a small area in the central nervous system that contains interconnected nuclei responsible for regulating food intake and energy expenditure. The major neuronal populations responsible for energy homeostasis are the orexigenic neuropeptide Y (NPY)/agouti-related peptide (AgRP) and the anorexic proopiomelanocortin (POMC)/cocaine and amphetamine-related peptide (CART) neurons [4,5]. These neurons are located in the arcuate nucleus (ARC) of the hypothalamus. This area is known to be highly accessible by circulating factors that can interfere with the central control of feeding and metabolism. For example, fatty acids can reach hypothalamic neurons and regulate neuronal signaling pathways by altering lipid membrane composition. The lipid profile in specific microdomains called lipid rafts (enriched in cholesterol, phospholipids, and sphingolipids) can interfere with several signaling pathways, such as insulin cascade, apoptosis, and autophagy [6]. Additionally, fatty acids sensed by neurons can also modulate neuronal activity by driving food intake [7,8,9]. Previous studies have demonstrated that a high-fat diet (HFD) or isolated saturated fatty acids (SFA), such as palmitate (PA), and stearic, arachidic, and behenic acids, are able to induce hypothalamic inflammation [10,11,12], which is associated with changes in *Npy* and/or *Pomc* expression and disruption of energy homeostasis control [12,13,14,15].

Excessive long-chain SFA (LCSFFA) can reach tissues, including the hypothalamus, and be channeled toward nonoxidative pathways, resulting in the production of toxic lipid species such as ceramide (Cer) and diacylglycerol, in a process termed lipotoxicity [16].

Thus, lipotoxicity is associated with inflammation, ER stress, and insulin/leptin resistance in a vicious cycle leading to hypothalamic dysfunction in which energy balance control is compromised.

More recently, sphingolipids have emerged as important regulators of energy homeostasis and inflammation in the hypothalamus. Sphingolipids are a family of lipids that play essential roles as structural cell membrane components and in cell signaling [17,18].

Sphingolipid metabolism is highly regulated by a complex network of interconnected pathways, with the major bioactive species including Cer, sphingosine, sphingosine-1-phosphate (S1P), and ceramide-1-phosphate (C1P). Specifically, ceramide is a key molecule in sphingolipid metabolism and a precursor of all other complex sphingolipids. Ceramides are widely distributed in cell membranes, where they play a crucial structural role and influence intracellular signaling by regulating physiological/pathological events such as cell proliferation, autophagy, apoptosis, and inflammation [12,19,20,21,22].

The biosynthesis of ceramide involves multiple metabolic pathways (Figure 1). One of these is de novo biosynthesis, which is initiated with the condensation of serine and palmitoyl-CoA by the enzyme serine palmitoyltransferase (SPT) in a rate-limiting step followed by several reactions involving key enzymes, such as 3-ketosphinganine reductase, ceramide synthase (CerS), and dihydroceramide desaturase (DES), resulting in ceramide production. In the context of FFA overload, de novo ceramide synthesis is mainly produced in the endoplasmic reticulum (ER) [20,22], but mitochondria also contribute to it because some enzymes involved in this process are also localized in this organelle. From the ER, ceramides are conducted through a specialized carrier system to the Golgi apparatus, where they can be phosphorylated by ceramide kinase into ceramide 1-phosphate (C1P), which is a bioactive sphingolipid involved in cell proliferation and inflammatory events [23,24]. Ceramide can also be produced through sphingolipid catabolism in a pathway called the “salvage pathway”, which involves the lysosomal degradation of sphingolipids into ceramide. The hydrolysis of sphingomyelin (SM) by acid or neutral sphingomyelinase (SMase) [20,25,26], the reacylation of sphingosine by ceramide synthases (CerS) [25,26], and the phosphorylation of sphingosine into S1P by sphingosine kinases [23] have the ability to produce the substrate for ceramide resynthesis. Depending on cellular metabolic status, glucosylceramide can produce ceramide by glucosylceramide synthase (GCS).

Sphingomyelinase can be induced for ceramide synthesis in response to specific biological factors, such as tumor necrosis factor alpha (TNF-α) or interleukin-1β (IL1-β) treatment [27,28], Fas ligand [29], oxidative stress [30], hormones [31,32,33], and excessive deposition of SFA [18], consequently enhancing ceramide levels.

Additionally, the inverse pathway can occur, leading to SM, sphingosine, and glucosylceramide production from ceramide by sphingomyelin synthase (SMS), ceramidase (CDase), and glucosylceramidase (GCase), respectively [24,25,26]. For example, the ceramide made in the endoplasmic reticulum (ER) can be introduced into the Golgi apparatus by its own carriers, subjected to different isoforms of the sphingomyelin synthase enzyme, and converted to sphingomyelin with a similar number of carbons to that of the original ceramide. This happens by converting phosphorylcholine to diacylglycerol [34,35]. The product from this reaction is mainly used for the manufacture of myelin of the cell membrane of neurons [36].

The length of the carbon chain of the added fatty acids for biosynthesis is variable and defines the species of ceramide formed. In mammals, the six existing ceramide synthases are responsible for the diversity of the species and tissue specificity of ceramides synthesized [37,38]. Depending on ceramide length and level, and the CerS isoform expressed, it could be associated with negative biological effects. For example, in peripheral organs/tissues, excessive ceramide has been correlated with lipotoxicity, thereby resulting in insulin resistance and β cell apoptosis [33]. It has been shown in vitro that insulin resistance in myoblasts induced by a saturated fatty acid such as palmitate is associated with increased levels of C16:0 ceramides (presence of a palmitic acid bound to sphingosine), whereas in vivo muscle insulin resistance was associated with ceramide C18:0 [39,40].

Although the nature of the ceramide species involved in insulin resistance due to obesity has not been fully elucidated, many studies have described the interactions of CerS isoforms in biological functions. Decreased CerS6 expression was associated with a protective effect from high-fat diet-induced obesity and glucose intolerance, and also increased energy expenditure and reduced C16:0 ceramide [41]. By comparison, CerS2 deficiency or insufficiency led to susceptibility to diet-induced steatohepatitis and insulin resistance, glucose intolerance, and increased C16 ceramide, suggesting that the altered sphingolipid acyl chain length directly affects insulin signaling, because it was observed as a compensatory effect for the decline in very-long-chain ceramide production by CerS2 [42,43].

Overall, excessive ceramide levels are related to decreased cell proliferation [44,45] and increased apoptosis [46,47].

Increased ceramide levels are also associated with macrophage infiltration after myocardial infarction, contributing to higher infarct size [48,49,50]. However, specific inhibition of SPT by myriocin leads to an approximately 35% reduction in the level of ceramides in the heart, which was associated with reduced infarct size [50].

More recently, it was reported that enzymes involved in de novo ceramide synthesis are widely expressed in the hypothalamus and hippocampus of rodents [51,52]. In an interesting review, Cruciani et al. (2017) [53] hypothesized that ceramide accumulation in the brain under lipotoxic conditions (triggered by HFD or isolated SFA treatment) might play a role in the downregulation of energy balance control, leading to obesity and associated diseases.

S1P is generated from sphingosine phosphorylation by the enzyme sphingosine kinase (Sphk), mostly in the plasma membrane. There are two isozymes of Sphk: sphingosine kinase 1 (Sphk1) and sphingosine kinase 2 (Sphk2) [54,55]. Sphk1 is found in the cytoplasm, whereas Sphk2 can be found in the nucleus [22] (Figure 1). The irreversible breakdown of S1P to phosphoethanolamine and hexadecenal is catalyzed by the enzyme S1P lyase (S1PL).

Interestingly, in contrast to ceramides and other metabolites, S1P has no structural role [20]. S1P mainly acts as a potent signal mediator that modulates multiple cellular functions important for health and diseases, usually with opposite effects of those of ceramide accumulation, and having tissue- or cell-type-specific effects. For example, Hadas et al. (2020) [56] used a new genetic tool called modified mRNA (modRNA) during myocardial infarction to increase the acid ceramidase (AC) enzyme, which increases S1P production, and found that the infarct area was reduced via decreasing cardiomyocytes’ cell death and infiltrated neutrophils. Interestingly, it has been found that feeding increases levels of S1P, showing a role in the regulation of energy balance [57,58]. Intracerebroventricular injections (ICV) of the bioactive lipid, S1P, reduce food consumption and increase rat energy expenditure through persistent activation of STAT3 and the melanocortin system.

There are five S1P receptor subtypes, with sphingosine-1-phosphate receptor 1 (S1PR1), sphingosine-1-phosphate receptor 2 (S1PR2), sphingosine-1-phosphate receptor 3 (S1PR3), sphingosine-1-phosphate receptor 5 (S1PR5), and, in some reports, sphingosine-1-phosphate receptor 4 (S1PR4) expressed in the central nervous system [59]. S1PR1 is highly expressed in regions of the hypothalamus, such as the arcuate and ventromedial/dorsomedial nuclei, which are involved in the control of feeding. S1PR3 is also expressed in astrocytes from rat and mouse brains [59].

Thus, sphingolipid metabolism has emerged as being important in energy balance control and is a pathway to be explored to better understand the pathophysiology of obesity. To date, few studies have addressed the relationship between fatty acids and sphingolipid metabolism in the control of energy homeostasis. Thus, our main objective was to evaluate the role of ceramide/sphingosine-1-phosphate and their influence on hypothalamic energy balance control, particularly in response to fatty acid interventions.

The results summarized here indicate that S1P, as opposed to ceramide, appears to be a central satiety factor in the hypothalamus. Extracted data demonstrated that ceramides are associated with hypothalamic dysfunction in which metaflammation, ER stress, and lipotoxicity lead to insulin/leptin resistance and disturbances in energy balance control, in a vicious cycle. We conclude that manipulation of sphingolipid levels would be an interesting strategy for clinicians to treat obesity, specifically by decreasing ceramide levels and the inflammation/ER-stress induced by overfeeding, particularly with LCSFA.

## 2. Materials and Methods

This review followed the recommendations of Preferred Reporting Items for Systematic Reviews and Meta-Analyses (PRISMA) guidelines. The review protocol was registered in PROSPERO International Prospective Register of Systematic Reviews (Unique Identifier: CRD42020198608).

### 2.1. Data Sources Search Strategy

Four electronic databases (Embase, PUBMED, Scopus, and Web of Science) were searched until May 2020, with an English language restriction. A new search was conducted during the final stage of the preparation of this article (January 2021). The search strategies were developed to trace published reports from studies with rodents and hypothalamic cells assessing the relationship between sphingolipids and energy balance control during obesity and associated conditions. Thus, the index terms used in our protocol were hypothalamus and sphingolipids.

The search was conducted by two authors, and the articles were selected by three independent reviewers. After the first screening, duplicates were removed from our databases. Initially, titles and summaries were screened. When full texts were available, we assessed them for eligibility according to our prespecified inclusion/exclusion criteria.

### 2.2. Eligibility Criteria

The population of interest was limited to rodents of any species receiving HFD, fatty acid treatment, or treatment with other obesogenic factors (such as inflammatory cytokines, sphingolipids), in addition to genetically obese mice models. All sexes and ages were considered. We also investigated studies that used hypothalamic cells treated with fatty acids, inflammatory cytokines, and/or sphingolipid metabolites (ceramides, S1P, or others). Our primary outcome included all data about the levels of sphingolipids. Our selected secondary outcomes were body weight, food intake, energy expenditure assessment, levels of neuropeptides, and inflammatory markers. We excluded articles that presented no criteria for population eligibility, intervention, comparison, and outcomes (Table 1).

### 2.3. Data Extraction

Data extraction was performed by two independent researchers after reading the previously selected articles, according to the eligibility criteria. A third researcher was consulted when there were differences or doubts regarding the article data. We extracted data including the following: names of authors, year of publication, species of animal, composition of diets, intervention period, neuropeptide levels, sphingolipid levels, inflammation levels, body weight, food intake, and adiposity. The data were transcribed into tables according to primary or secondary outcomes, separating animal results from cellular models.

### 2.4. Assessment of Methodology Quality

The evaluation of risk of bias in the studies included in this systematic review was conducted in pairs and individually for each article. For in vivo studies, assessment was based on Syrcle’s recommendations [60]. This tool establishes 10 questions related to selection bias (sequence generation, baseline characteristics, and allocation concealment), performance bias (random housing and blinding researchers), detections bias (random outcome data and blinding of outcome assessment), incomplete outcome data (attrition bias), selective outcome reporting (reporting bias), and other biases.

For in vitro studies, we adapted Syrcle’s tool, based on the studies of Golbach et al. (2016) [61] and Alshwaimi et al. (2016) [62]. Thus, we applied 10 questions to the analyzed articles, individually. The questions were related to selection bias (baseline characteristics of cell culture models, duration and frequency of the exposure, and cellular viability), performance bias (controlled temperature and humidity, randomized and independent experiments), detection bias (blinding), incomplete outcome data (attrition bias), selective outcome reporting (reporting bias), and other biases. These questions were judged as having high, uncertain, or low risks of bias. In Figure 2 and Figure 3, the green symbol means a high level of reliability, the yellow symbol means not clear, and the red symbol means low reliability.

## 3. Results

The main aim of this systematic review was to evaluate ceramide/S1P and their influence on hypothalamic energy balance control, particularly in response to fatty acid interventions.

An initial search in the electronic databases (Embase, PUBMED, Scopus, and Web of Science) returned a total of 269 articles. The titles and abstracts of 31 articles were screened. Subsequently, nine duplicates were removed by the authors. We also excluded six full-text articles according to the exclusion criteria (Table 1 and Figure 4). All articles found were published in English. At the final stage of the writing process, we conducted a new search (January 2021) and included three more articles, for a total of 24 studies [15,51,57,58,63,64,65,66,67,68,69,70,71,72,73,74,75,76,77,78,79,80,81,82]. These steps are described in the flowchart presented in Figure 4.

## 4. Methodological Approach to Assess Sphingolipids

After article selection, the first step was analysis of the manner in which researchers assessed sphingolipid levels and enzymes involved in their synthesis. First, we observed that in vitro studies (Table 2) used the following methods: commercial kits (n = 1); lipidomics (n = 1); liquid chromatography coupled with mass spectrometry (n = 2); liquid chromatography coupled with electrospray ionization and mass spectrometry (n = 1); mass spectrometry (n = 1). Studies also used Western blot (n = 1) and quantitative polymerase chain reaction (qPCR) (n = 1) to monitor enzymes involved in the pathway. However, it should be noted that these are not suitable methods for ceramide quantification. Finally, some papers had no information on the methodological approach used or did not present sphingolipid levels (n = 3), as described in Table 2.

By analyzing in vivo data (Table 3 and Table 4), we observed that the studies used lipidomics (n = 1), liquid chromatography (n = 2); mass spectrometry (n = 1); liquid chromatography coupled with mass spectrometry (n = 7); electrospray ionization (n = 1); liquid chromatography coupled with electrospray ionization and mass spectrometry (n = 2); electrospray ionization coupled with mass spectrometry (n = 1); Western blot and qPCR (n = 1); RNASeq associated with qPCR (n = 1); and no assessment or no information on methodology (n = 1) (Table 4).

## 5. Additional in Vitro Methodological Approach

As mentioned above, we decided to include in vivo research and in vitro findings in our review because the anatomy of the hypothalamus is highly complex, and mechanistic investigations of neuropeptide gene regulation and signal transduction events occurring in neurons are a challenge to perform in vivo [14]. Thus, some of our selected studies used hypothalamic cells (primary or lineage) alone or combined with animal models (n = 10 studies), as shown in Table 2. In general, these cells were exposed to fatty acids (palmitate, oleate, lauric, oleanolic, linoleic, or stearic acid) and had ceramide levels assessed (n = 8 studies) (Table 2). Other treatments, such as TNF-α, IL1-β, estradiol, leptin, lipopolysaccharide (LPS), or Toll-like receptor 4 (TLR4) inhibitor, were also applied in these studies (n = 10) (Table 2).

## 6. Main Findings Regarding In Vitro Sphingolipid Levels and Their Role in the Hypothalamus

Interestingly, as previously reported in cells from peripheral tissues [48,49,83], the studies selected here identified a relationship between ceramide levels and pro-inflammatory cytokines in hypothalamic cells. Importantly, these levels are tissue- and stimulus-specific and dependent on the type of ceramide species assessed. For example, the authors of [77] found that exogenous C6 ceramide treatment in microglial cells decreased interleukin-6 (IL-6) levels (Table 2). In another study, Dusaban et al. (2017) [80] found that S1PR3, but not S1PR1, induces an increase in IL-6, vascular endothelial growth factor A (VEGFa), and cyclooxygenase-2 (COX-2) mRNA levels in mice astrocytes (Table 2). Interestingly, the authors also found that astrocyte inflammation in vitro leads to an increase in Sphk1 and S1P3 expression, which they speculated could contribute to astrocyte autocrine inflammatory signaling.

Furthermore, treatment with exogenous TNF-α or IL1-β was able to increase neuronal ceramide production that was associated with neuronal apoptosis [74,75] (Table 2). Sortino et al. (1999) [75] also investigated the effects of decreasing C1P levels using D609 (tricyclodecan-9-xanthogenate), an inhibitor of phosphatidylcholine-specific phospholipase C. This is a key enzyme in the activation of Ac SMase (sphingomyelin synthase) responsible for transferring the phosphocholine group from phosphatidylcholine to ceramide-generating sphingomyelin and diacylglycerol (DAG) (Figure 3). Using this approach, the authors found that D609 decreased the accumulation of C1P in response to 15 min exposure to 20 ng/mL TNF-α by 50% (Table 2), and partially prevented TNF-α-induced apoptosis [75].

PA is also associated with increased ceramide levels and neuronal inflammation [63,67,79] (Table 2). Morselli et al. (2014) [68] showed that PA decreases peroxisome proliferator-activated receptor gamma coactivator-1 alpha (PGC-1α) and estrogen receptor (ERα) expression in hypothalamic neurons and astrocytes, thereby promoting inflammation associated with increased ceramide levels (Table 2). Consistently, Sergi et al. (2018) [79] showed that L-cycloserine (a SPT inhibitor, Figure 1) inhibited PA-induced inflammation in the hypothalamic mHypoE-N42 cell line (Table 2). Conversely, oleic acid (OA) and eicosapentaenoic acid (EPA) exerted anti-inflammatory effects by decreasing PA-induced intracellular ceramide build-up (Table 2). The authors also tested the involvement of the TLR4 receptor on neuronal inflammation in response to fatty acids and unexpectedly found that TLR4 inhibitors failed to inhibit PA-induced upregulation of proinflammatory cytokines.

The in vitro studies, although providing potential mechanistic insights into the role of ceramide species in the regulation of energy balance, lack the ability of in vivo studies to provide a physiological outcome, such as food intake. In this context, Silva et al. (2014) [57] showed that S1P treatment increases the signal transducer and activator of transcription 3 (STAT3)-dependent axis in GT1-7 cells (Table 2). By comparison, Tse et al. (2018) [15], using hypothalamic mHypoA-POMC/GFP neurons, found that PA increases neuronal Pomc expression. This phenomenon is dependent on palmitoyl-CoA synthesis, but not de novo ceramide synthesis, since inhibition of SPT enhanced palmitate-induced Pomc expression (Table 2). McFadden et al. (2014) [70] found that exposure to palmitate promotes lipidomic remodeling in primary hypothalamic neurons (PHNs). The authors observed an increase in lipid species, including ceramides 18:0/16:0, when PHN neurons were exposed to excess C16:0. This outcome was reversed by increasing FA catabolism with C75, a stimulator of carnitine palmitoyltransferase-1 (CPT1C) and fatty acid oxidation. This stimulation of FA oxidation in neurons produced an increase in ATP levels, which the authors suggested led to a decrease in food intake and body weight. Finally, Campana et al. (2017) [63] demonstrated that myriocin treatment, in addition to serine palmitoyltransferase 2 (SPT2) knockdown by small interfering RNA (siRNA) (Figure 1), was able to restore neuronal AKT phosphorylation [63] in neuronal GT1-7 cells, which is possibly associated with improved energy balance control under obesogenic conditions.

In summary, these studies demonstrated that saturated fatty acids and inflammatory cytokines are closely related to the sphingolipid pathways, thereby influencing food intake, which is of potential importance for the understanding and treatment of obesity.

## 7. Additional In Vivo Methodological Approach

We found 24 articles in total with n = 18 using animals to answer their main hypotheses, as shown in Table 3 and Table 4. The strains used were Ob/Ob mice (n = 1); Db/Db mice (n = 1); Swiss mice (n = 1); C57BL/GJ mice (n = 7); Sprague Dawley rats (n = 4); Zucker rats (n = 2); Wistar rats (n = 4); and models of knockout mice (n = 4) (Table 3). The male sex was the most used (with n = 12 only using male) (Table 3). Only two studies (n = 2) used females exclusively (Table 3). Finally, some studies used both (n = 4), as shown in Table 3. In general, the experimental age started at 5 weeks, with some studies analyzing animals up to 10 weeks old (Table 3).

Regarding the stimulus used to examine differences between sphingolipid levels, we mainly focused on studies administering HFD and/or FFA treatment. Other treatments/manipulations were reported (such as hormones, exogenous ceramide, agonists, or antagonists of the sphingolipid pathway). We used these manipulations to compare or improve our conclusions regarding diet exposure. We classified the intervention into three classes: hormones/genetic tools (n = 3); HFD or FFA exclusively (n = 4); and mixed stimulus (HFD or FFA combined with hormones, genetic tools, feeding/refeeding, exogenous ceramide treatment, overnutrition by litter size reduction, and others), as shown in Table 3.

## 8. Main Findings Regarding In Vivo Sphingolipid Levels and Their Role in the Hypothalamus

Growing evidence suggests that hypothalamic lipid sensing plays a key role in controlling energy balance [7,84], and that its dysregulation could lead to the development of obesity and associated comorbidities, such as insulin resistance and type 2 diabetes [7,10,57].

Interestingly, it has been discussed in the literature that there is a sexual dimorphism in brain metabolism in response to HFD. Following HFD consumption, the brain tissue of males, but not females, showed elevated levels of saturated fatty acids, such as PA [67,68]. The authors also found that male mice presented elevations in ceramides, which were associated with increased markers of inflammation when compared to female mice (Table 4). They also found that recovering ERα expression in males was able to reduce markers of inflammation [67,68].

Gonzales-Garcia (2018) [65] consistently found that central steroid hormone 17-β estradiol (E2) treatment decreases hypothalamic ceramide levels (Table 4) and ER stress, and increases brown adipose tissue (BAT) thermogenesis (Table 3). Increased ceramide levels were also associated with obesity and earlier female puberty in a study by Heras et al. (2020) [81] (Table 3 and Table 4). However, when the authors blocked CerS, the obese phenotype was delayed [81] (Table 3 and Table 4). Thus, Cer metabolism and ER stress pathway components could be potential therapeutic targets for the treatment of obesity associated with irregular estrogen levels. It is also important to mention that ceramide is able to trigger ER stress in the hypothalamus, impacting the energy balance [51].

HFD feeding was associated with increased ceramide levels and inflammation by other studies reported here using male animal models [72,78,82] (Table 3 and Table 4). For example, obese male mice also presented increased hypothalamic ceramide levels [51,63,81] (Table 4). Overall, some authors suggested that lipid content in the hypothalamus is directly determined by dietary lipid content and not obesity per se, because exercise training for 6 weeks ameliorated inflammation triggered by diet but did not affect hypothalamic lipid profile [72] (Table 3 and Table 4). Notably, Campana et al. (2018) [63], using deuterated palmitic acid, demonstrated that, in the hypothalamus, PA activates several enzymes of the de novo ceramide synthesis pathway, which is associated with the development of insulin resistance. Interestingly, ICV myriocin treatment reduced hypothalamic ceramide levels (Table 4) and improved insulin sensitivity. However, it is important to note that body weight reduction can also lead to higher ceramide levels in some areas (plasma, liver, and adipose tissue) [78].

Lyu et al. (2020) [82], using transcriptomic analysis from animals receiving HFD, showed upregulation of genes related to sphingolipid metabolism associated with loss of POMC neurons (Table 4). Interestingly, treatment with telmisartan recovers HFD-induced ceramide levels, preventing lipotoxicity and contributing to weight loss (Rawish et al., 2020) (Table 3). Thus, lipid intake appears to be critical to sphingolipid metabolism and neuropeptide expression, confirming our main hypothesis.

Ceramide levels in the brain are also associated with lipoprotein lipase (LPL) activity [66]. LPL deficiency is associated with increased ceramide levels in the hypothalamus, in addition to increased body weight (Table 4) and glucose intolerance. The authors showed that LPL is differentially regulated by saturated and unsaturated fatty acids, and the lack of LPL causes ceramide accumulation that is associated with microglia inflammation, ER stress, and increased AgRP levels (Table 4) [66]. Laperrousaz et al. (2017) [64] also found that the partial LPL in mediobasal hypothalamus (MBH) increases body weight and decreases locomotor activity, without changes in AgRP/NPY levels. However, the authors found that decreased LPL in MBH led to a lack of ceramides. It is possible that the differences between the protocols and brain area analyzed are the main reason for the different results found by Gao et al. (2017) and Laperrousaz et al. (2017) [64,66].

Others have addressed the association between ceramides and food intake. For example, ceramide administration was able to induce food intake and orexigenic neuropeptide expression (NPY and AgRP) in CPT1C knockout (CPT1C KO) mice [73] (Table 3 and Table 4). The authors suggest a ghrelin/CPT1C/ceramide axis is important for the understanding and treatment of obesity [73], in which CPT1C KO mice had the orexigenic action of ghrelin totally abolished. The same phenotype was observed with myriocin treatment. Others have noted an important relationship between ceramides and CPT1C. For example, Gao et al. (2011) [71] found that leptin decreases CER through malonyl-Coa and, consequently, CPT1C, which in turn affects NPY expression and food intake (Table 3 and Table 4). Moreover, Mera et al. (2014) [69] demonstrated that CPT1C contributes to feeding regulation by modulating neurotransmitter expression and neuronal lipid profiles (Table 3 and Table 4).

Contreras et al. (2014) [51] also showed that ceramides regulate energy balance through the induction of hypothalamic inflammation, and ER stress associated with reduced BAT thermogenesis and weight gain (Table 3 and Table 4). Maldonado et al. (2019) [77] also found increased hypothalamic inflammation (Table 4).

Finally, Silva et al. (2014) [57] found that S1P ICV treatment decreases food intake and increases energy expenditure (Table 3) by increasing STAT3-phosphorylation and POMC expression in lean mice. Importantly, using obese mice, the authors found decreased levels of S1PR1, whereas ICV S1P treatment decreased food intake, body weight, and adiposity (Table 3 and Table 4). Vozzella et al. (2019) [58] also consistently found that feeding increases S1P, whereas fasting decreases it. This modulation is related to the de novo pathway.

In summary, the studies here provide evidence that S1P, as opposed to ceramide, appears to be a central satiety factor in the hypothalamus. A critical examination of the use of WB and RTq-PCR must always be undertaken to interpret data from the literature, because these methods are not suitable for ceramide quantification. Finally, sphingolipid assessment could be an important pathway to explore to help the scientific community develop new therapies for patients with obesity and to treat their metabolic disorders.

## 9. Discussion

In recent years, numerous studies have noted that obesity and the excessive consumption of SFA are accompanied by hypothalamic dysfunction driven by lipotoxicity, ER stress, metaflammation, oxidative stress, autophagy, and apoptosis. As a result, neurons and glial cells show impaired function, resulting in disorders in food intake and energy expenditure. The hypothalamus integrates the signals from the periphery to transmit signals within the brain and back to the peripheral organs to sustain energy balance. The energy derived from neuronal mitochondria is crucial in this context [85], and dysfunction in this organelle is related to changes in homeostasis observed in obesity. Indeed, neuronal mitochondrial dysfunction has been linked to the higher SFA content found in lard, as the fat source of HFD, because PA treatment of mHypoA-2/23 CLU183 cells leads to mitochondrial fission in addition to altered mitochondrial dynamics, suggesting that SFAs affect mitochondrial activity [86].

Increased evidence has shown that, as in the case of SFA, dietary intake of an increased ω6- to ω3-polyunsaturated fatty acid (PUFA) ratio is correlated with metabolic diseases [87]. A Western-type diet containing soybean oil and cholesterol caused obesity in mice with alterations in liver function similar to those observed during metabolic syndrome in humans, including mitochondrial dysfunction [88]. Although linoleic acid (LA) is considered to be beneficial for mitochondrial activity, due to its incorporation in cardiolipin species, the in vitro treatment of hypothalamic cells with LA induces a rise in protein carbonylation, promoting oxidative damage in neurons [86].

Preventing obesity is a major challenge for clinicians and researchers because this disease has diverse causes, including genetic and epigenetic factors associated with lifestyle. Similarly, understanding the effects of not only the amount of fat ingested, but also the type of dietary fatty acids, on energy homeostasis under different conditions represents a challenge.

Here, we highlighted the consequences of hypercaloric intake, including excessive sugar and saturated fatty acid intake, usually from ultraprocessed food (UPF) on the dysmetabolic phenotype. UPF is associated with increased risk of cardiovascular diseases and all causes of mortality [89]. Even more concerning is the fact that children under 3 years of age eating UPF showed increased blood cholesterol levels at 6 years of age [90]. Therefore, it is highly important to understand the alteration of pathways in response to excessive SFA intake. This review aimed to address this important issue by examining the available evidence on the involvement of fatty acids and inflammation in ceramide levels and their consequences for hypothalamic function, particularly related to energy balance control.

It should be noted that sphingolipid species possess substantial structural diversity in contrast to the common ceramide structure, and as a result, their biological effects differ. Ideally, therefore, the methodological approaches used to study sphingolipids must present a high degree of specificity, distinguishing the structural similarity and interconversion between samples, thereby providing an accurate determination of metabolic flux [91].

Here, we found that some studies used common molecular biology tools, such as Western blot and qPCR to monitor enzymes, receptors, or genes involved in the process. These are easier and cheaper methods used in all molecular biology laboratories. However, it should be noted that these methods are not validated for assessing ceramide levels. Gene expression or protein content (for example, CerS and Sphk) cannot be always correlated with sphingolipid levels involved in the process because post-translational regulation may occur. Therefore, the optimal approach to validate gene and protein expression data is via a direct assessment of ceramide levels using analytical techniques such as high-performance liquid chromatography–mass and tandem mass spectrometry (HPLC-MS/MS). Additionally, pharmacological tools such as classical agonists/antagonists, in addition to genetic manipulations such as viral injection or knockout models, are useful tools for helping to interpret the data from Western blots and qPCR.

High-performance liquid chromatography (HPLC) is a relatively inexpensive tool that is able to provide a better resolution of sphingolipid levels. This approach has the capability to trace, quantify, and determine the structures of free sphingoid bases, free sphingoid base phosphates, ceramides, and sphingomyelin, among others, in a single sample, showing molecular mass and structure [91]. This technique provides high sensitivity by detecting low magnitudes, and the signal from the samples can be correlated with their concentration, constituting quantitative data that researchers can use to better understand biological effects of several stimuli [91,92]. However, accurate quantification usually depends on the definitive resolution of sphingolipid species, which HPLC alone cannot guarantee. This disadvantage exemplifies the requirement of additional techniques for structural elucidation [91].

For example, additional approaches found in the selected studies were electron ionization mass spectrometry (EI) and electrospray ionization mass spectrometry (ESI). EI enables the analysis of intact sphingolipid species, in addition to fragmented molecular species, whereas ESI enables sphingolipids in solution to be continuously infused at a low flow rate, providing great sensitivity. Finally, we also observed that HPLC-MS/MS is usually the analytical tool of choice for “sphingolipidomic” studies [92].

Here, our main focus was related to the description of recent studies regarding hypothalamic energy balance control and obesity, particularly triggered by HFD, because it has been widely documented that overnutrition, especially lipid overload, induces a state of hypothalamic dysfunction in which low-grade inflammation (or “metaflammation”) through TLR4 receptor and NFK-B pathways, ER stress, and lipotoxicity are tightly associated with insulin/leptin resistance in a vicious cycle.

Thus, obesity has been associated with increased de novo ceramide synthesis in response to a higher bioavailability of FFA, particularly LCSFA, such as palmitate [72,93]. Recent work involving acute lipid infusions in rats suggests that, in situations of acute lipid oversupply, ceramide levels particularly increase if the FA source provided is saturated [94], most likely because this promotes de novo ceramide synthesis through the incorporation of palmitate at an early rate-limiting step related to an increase in palmitoyl-CoA [95]. Thus, palmitoyl-CoA formation and TLR4 stimulation, which has been previously related to the activation of SMases that convert SM to CER (Figure 3), are responsible for increased Cer levels in response to lipid overload [96,97]. As we described in our summarized results, others have found that levels of proinflammatory cytokines, such as IL-6 and TNF-α, can upregulate SMases and SPT (Figure 3), triggering excessive Cer levels [32]. Interestingly, n-6 PUFA consumption was also associated with increased ceramide levels in muscle (C16:0, C18:0; C22:0, and C24:1) [98]. Furthermore, when ceramide synthesis was blocked using L-cycloserine, there was an observed decrease in PA-induced inflammation [79].

In contrast, unsaturated fatty acids have been reported to improve insulin sensitivity and metaflammation. This may be because LCSFAs are less prone to β-oxidation when compared to medium-chain fatty acids and unsaturated fatty acids [16]. Importantly, when reviewing the literature, we found a lack of in vitro and in vivo studies showing the assessment of ceramide levels in response to unsaturated fatty acids such as palmitoleate or EPA/DHA. For instance, the authors of [99] used GLP-1-secreting GLUTag cells and found that PA (C16:0) but not palmitoleate (PO, C16:1) induces ceramide production, whereas unsaturated fatty acids confer lipoprotection by enhancing cell viability. Additionally, unsaturated fatty acids can ameliorate inflammation triggered by palmitate [79,100,101]; therefore, it is possible that they can decrease ceramide levels or at least reverse the effects of ceramide. However, further studies are needed to clarify this relationship.

Fatty acid metabolism is also an important regulator of cellular Cer levels and hypothalamic energy balance control by the CPT1-depending axis. Lipid overload or an excess of energetic substrate in the cell (such as carbohydrates) increases the ATP:ADP ratio, promoting citrate efflux from mitochondria into the cytoplasm and inducing formation of malonyl-CoA, which classically inhibits CPT-1. The inhibition of FA transport to mitochondria increases the availability of palmitoyl-CoA in the ER for de novo ceramide synthesis, and it appears to be one of the main pathways for ceramide synthesis in obesity. The brain mainly contains a specific CPT1 isoform, named CPT1C, which is localized in the ER of neurons. CPT1C was the last member of the CPT1 family of genes to be described, whereas CPT1A is present in the liver and other tissues capable of high rates of FA synthesis, and CPT1B are expressed mainly in tissues characterized by high rates of fatty acid oxidation, such as muscle and brown adipose tissue [102]. CPT1C has been proposed to be a sensor of malonyl-CoA levels in hypothalamic neurons, therefore contributing to energy homeostasis. Indeed, CPT1C KO mice have reduced food intake and weight gain [103]. Additionally, Ramirez and colleagues (2013) [73] found that ceramide metabolism is involved in the anorectic effects of CPT1C in the Arc. The authors discovered that CPT1C KO mice are unresponsive to orexigenic effects of ghrelin. Interestingly, ghrelin seems to act in an additional pathway, increasing levels of C18:0 ceramide, which is mediated by CPT1C. Accordingly, myriocin completely blunted the orexigenic action of ghrelin. Overall, these data indicate that, in addition to formerly reported mechanisms, ghrelin also induces food intake through the regulation of hypothalamic CPT1C and ceramide metabolism, a finding of potential importance for the understanding and treatment of obesity.

Because CPT1C and SPT are expressed in the endoplasmic reticulum, previous studies suggest that palmitoyl CoA might be available via CPT1C action, which induces de novo ceramide synthesis. Additionally, it is possible that CPT1C can also act as a transporter of palmitoyl-CoA to the endoplasmic reticulum [17]. Moreover, it was shown that the stimulation of FA oxidation in neurons produced an increase in ATP levels, leading to a decreased food intake and body weight [70]. The authors showed that an increase in FA catabolism through CPT1C stimulation reversed the lipidomic remodeling in PHN hypothalamic neurons promoted by palmitate. Thus, FA metabolism by CPT1C seems to constitute an additional mechanism that regulates ceramide levels; however, controversy remains regarding these findings.

Because CPT1C is present in the ER, it does not participate in β-oxidation. Mera et al. (2014) [69] found that CPT1A overexpression in the ventromedial hypothalamus contributes to an obese phenotype, characterized by hyperphagia and, consequently, overweight, insulin resistance, glucose intolerance, and hyperglycemia, and led to higher levels of ceramides and sphingolipids, accompanied by increased mitochondrial reactive oxygen species, demonstrating an association between mitochondrial function and ceramide levels. Contributing to this evidence, Jeng and colleagues (2009) [104] found that higher ceramide levels led to mitochondrial dysfunctions, which alter fatty acid catabolism. They treated C6 glioma cells with C2 ceramide (50uM) and observed an increased cytochrome c release and caspase-3 activation, which could trigger apoptosis. The authors hypothesize that this ceramide-induced cell death may have been due to disruption of mitochondrial membrane potential (analyzed with 5,5′,6,6′-tetrachloro-1,1′,3,3′- tetraethylbenzimidazolylcarbocyanine iodide JC-1), ATP production, and mitochondrial biogenesis. Stoica et al. (2002) [105] showed a rapid mitochondrial depolarization and activation of the intrinsic caspase pathway, as determined by cytochrome c release to cytosol and increased caspase-9 content in primary rat cortical neurons with C2 ceramide (50 uM) treatment. As mentioned above, these events may induce FA oxidation disturbances because they occur in mitochondria compartments.

Furthermore, HFD feeding increases the total content of several neutral lipid species, such as phospholipids and triacylglycerol, while also increasing signaling lipids, such as sphingolipids and diacylglycerol, among others. Interestingly, fatty acid profiles appear to have an important role in sphingolipids, because mice subjected to 6 weeks of training showed no reduction in ceramide levels, despite having reduced inflammation. Thus, obesity per se does not seem to be the most important stimulus for altering hypothalamic lipid levels [72]. In addition, recent findings showed the existence of sexual dimorphism related to metabolism and obesity. For example, male mice had elevated levels of saturated fatty acids, such as PA, and elevations in total sphingolipids and ceramides, compared to female mice. Such changes were associated with elevated markers of inflammation in male mice [67,68], emphasizing the importance of sex-based research.

Overall, the literature reveals an inverse relationship between ceramide levels and insulin sensitivity in both rodents and humans [106]. For example, excessive ceramide levels were found in both the plasma and skeletal muscle of diabetic patients, and in the liver, plasma, and muscle of obese mice [107,108,109,110]. Studies in rodents show that inhibiting ceramide biosynthesis also warded off several pathologies associated with insulin resistance, including diabetes, atherosclerosis, hepatic steatosis, and cardiomyopathy. Mechanistically, in peripheral tissues, in addition to the relationship with metaflammation, the additional effects of ceramides on insulin resistance appear to result from their ability to block activation of Akt/PKB, a serine/-threonine kinase that is necessary for growth-factor activation of anabolism, the metabolic effects of insulin, and cell survival [106]. Additionally, ceramides have a detrimental effect on pancreatic β cells, by activating the stress-induced apoptotic pathway (i.e., cytochrome C release and free radical production) [111].

In the hypothalamus of rodents, lipid infusions or high-fat feeding are able to increase ceramide levels [97]. Obese Zucker rats also have higher levels of ceramide in the hypothalamus [63]. Mechanistically, PA treatment impairs insulin signaling while increasing ceramide levels in hypothalamic neuronal GT1-7 cells, whereas myriocin or SPT2 siRNA counteracts PA-induced insulin resistance in these cells [63]. Using an in vivo model (obese Zucker rats), myriocin treatment partially restored glucose-homeostasis.

Ceramides stimulate activation of atypical PKC isoforms (PKCζ/λ), which favors their combination with PKB/Akt and further prevents the activation of PKB/Akt in response to insulin [83], contributing to the insulin resistance state. In fact, PKCζ is a highly expressed kinase in the arcuate and paraventricular nuclei from the hypothalamus [112,113]. Interestingly, Campana and colleagues (2018) demonstrated a positive impact of the PKC inhibitor on insulin resistance induced by obesity. They found that the dominant negative form of PKCζ completely prevented hypothalamic insulin resistance induced by ceramide or palmitate. In addition to the central nervous system, PKCζ was also related to peripheral insulin resistance [83], suggesting a unique role of this PKC isoform in the regulation of insulin sensitivity by ceramides.

Thus, de novo ceramide synthesis has a key role in hypothalamic insulin resistance development and glucose homeostasis dysregulation associated with obesity. Here, we also summarized that obesity and associated conditions, such as metaflammation and ER stress, are able to increase hypothalamic ceramide levels, which in turn also have a role in triggering these pathways and inducing insulin resistance.

For example, exogenous ceramides can induce hypothalamic dysfunction by triggering ER stress. C6 ceramide treatment leads to weight gain and increased C16 levels in the mediobasal hypothalamus, which is reversed by GRP78 overexpression [51]. GPR78 is a chaperone that improves protein folding and, when administrated in obese Zucker rats, leads to reduced body weight, and increases thermogenesis and insulin/leptin signaling. However, the levels of ceramide remained elevated when compared to the control group [114], perhaps indicating that ER stress is downstream of the effects of ceramide [114].

This evidence confirmed our initial hypothesis that recent studies have provided information regarding the negative effects of ceramides on hypothalamic function [63]. Other examples reinforcing this idea have used transgenic mice. For example, LPL knockout mice exhibit excessive ceramide levels, which in turn trigger microglial activation and increased AgRP expression levels [66]. This phenomenon was associated with increased feeding, weight gain, and adiposity [51,81]. Recently, Lyu et al. (2020) [82] also noted that irregular sphingolipid metabolism pathways may be associated with the loss of POMC neurons under HFD feeding.

In contrast to the increasing evidence supporting the role of ceramides in obesity and associated disorders, S1P is a recently discovered molecule that appears to have positive effects on metabolism. Specifically, the ICV administration of S1P can decrease food intake and increase energy expenditure with promising results, reversing obese phenotypes in mice. The same effect was observed when a S1P agonist (SEW2871) was administered; mice had an induced anorexigenic effect, preventing obesity and associated metabolic diseases in a STAT3/POMC-dependent axis [57].

Interestingly, once outside the cell, S1P can either bind to albumin [115] or apolipoprotein M (ApoM) [116]. The ApoM/HDL-bound S1P has been found to be a contributor to the vasoprotective effects of HDLs [117], such as antiatherogenic properties [118]. However, the relationship of S1P with insulin resistance and type 2 diabetes (T2D) remains controversial in some tissues. For example, S1P inhibited insulin signaling in the liver, both in vitro and in vivo [119], and PA increases ceramide and S1P in beta-cells [120].

Thus, we highlight that the understanding of sphingolipid metabolism is a prospective area of interest because ceramides/S1P can be targeted to monitor obesity (as a biomarker) or to treat it using pharmacological nutritional approaches. However, precise determinations of different ceramide subspecies require further elucidation, thus warranting the characterization of their actions in obesity and related comorbidities, particularly regarding tissue-specific effects. In addition, exploring the status of sphingolipid pathways in models of metabolic reprogramming by HFD can be a useful tool to understand the obese phenotype, particularly in young individuals.

## 10. Conclusions

We conclude that our work contributes to summarizing the evidence related to sphingolipid metabolism and its role in hypothalamic energy balance control. The studies analyzed here monitored sphingolipid levels using different methodological approaches; however, it should be noted that RTq-PCR and WB are not suitable methods for ceramide quantification, and the results need additional interpretation.

Regarding energy homeostasis, ceramide accumulation in the hypothalamus causes inflammation, ER stress, and insulin/leptin resistance, thus interrupting the energy balance associated with an obese phenotype. Conversely, the S1P axis shows the opposite effects on body weight. Dietary approaches appear to be an interesting management strategy because recent evidence has shown that lipids, particularly saturated fatty acids such as PA, can interfere with de novo ceramide synthesis. It should be noted that other models have found an exacerbated hypothalamic injury in response to excessive consumption of SFA associated with dietary simple sugars. Thus, manipulation of sphingolipid levels with drugs or nutrients can be useful approaches for the treatment of obesity by clinicians, particularly by aiming to decrease ceramide levels to have a positive effect on insulin/leptin resistance. Further studies to investigate specific ceramide chains and the impact of unsaturated fatty acids are needed to clarify their specific action on metabolism.

## Figures and Tables

**Figure 1 ijms-22-05357-f001:**
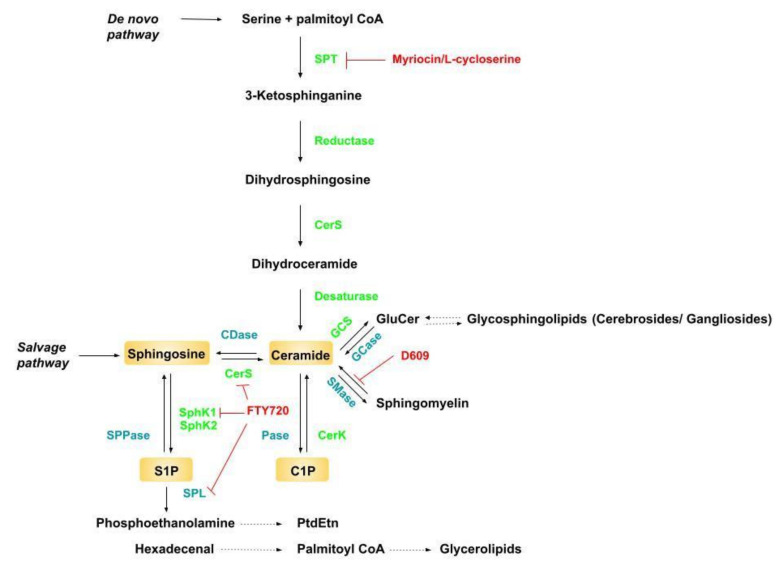
Sphingolipid general pathway. Enzymes are presented in green and drugs commonly used in the studies reviewed here are presented in red. CDase—Ceramidase; CerK—Ceramide Kinase; CerS—Ceramide Synthase; C1P—Ceramide-1-Phosphate; D609—Tricyclodecan-9-yl-Xanthogenate; FTY720—Fingolimod; GCase—Glucosylceramidase; GCS—Glucosylceramide Synthase; Pase—Phosphatase; PtdEtn—Phosphatidylethanolamine; SMase—Sphingomyelinase; SphK—Sphingosine Kinase (1 and 2); SPL—Sphingosine-1-phosphate Liase; SPPase—Sphingosine Phosphate Phosphatase; SPT—Serine Palmitoil-CoA Transferase; S1P—Sphingosine-1-phosphate.

**Figure 2 ijms-22-05357-f002:**
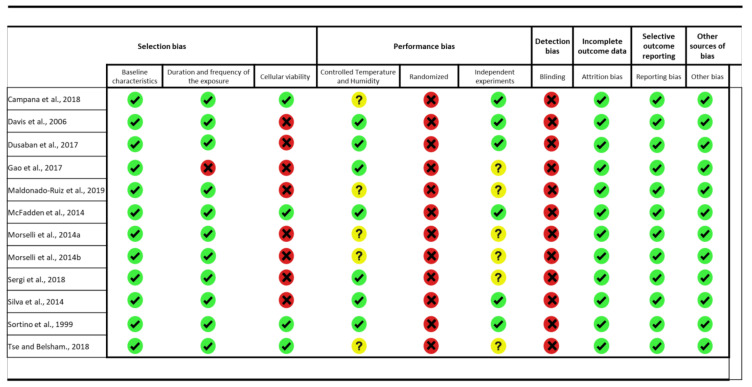
Risk of bias (RoB) from in vitro studies. Summary of the authors’ judgments concerning risks from included in vitro studies. 
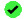
 = high reliability, 
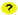
 = not clear, 
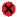
 = low reliability.

**Figure 3 ijms-22-05357-f003:**
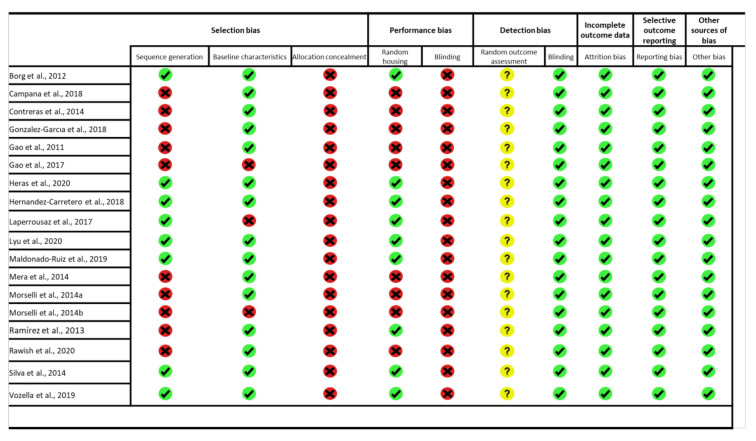
Risk of bias (RoB) from in vivo studies. Summary of the authors’ judgments concerning risks from included in vivo studies. 
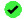
 = high reliability, 
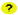
 = not clear, 
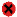
 = low reliability.

**Figure 4 ijms-22-05357-f004:**
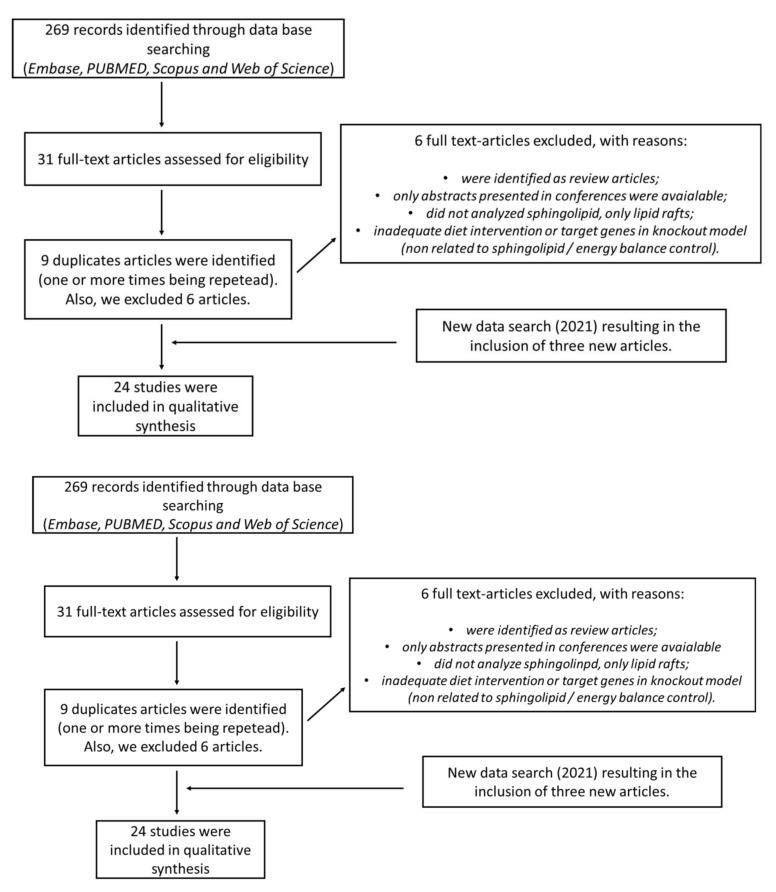
Flowchart of study selection process applied in this study.

**Table 1 ijms-22-05357-t001:** Eligibility criteria applied in this study.

	Inclusion Criteria	Exclusion Criteria
Population	Rodents (offspring and/or adults); hypothalamic cell lines; or primary hypothalamic cell culture.	Other cells than hypothalamic; non-rodents and studies involving human subjects.
Intervention	High-fat feeding; genetic models of obesity; treatment with fatty acids; treatment with cytokines; treatment with sphingolipids (ceramide or sphingosine-1-phosphate).	Other genetic backgrounds; animals with protein restriction or receiving any other treatment.
Comparison	Sphingolipid levels (ceramide and sphingosine-1-phosphate); fasting.	Studies related to ganglioside metabolism.
Outcomes	Alterations in sphingolipid metabolism following disturbances in energy balance control. Primary outcomes: sphingolipid level (ceramide and sphingosine -1-phosphate). Secondary outcomes: Body weight, food intake, energy expenditure assessments, neuropeptides levels, and inflammatory markers.	Other outcomes not relevant to energy balance control.
Type of publication	Original article	Nonoriginal article (reviews, conference abstracts).
Other selection criteria	Studies following these main characteristics were included: animal and cell culture studies from 1990 until 2020. The articles needed to be published on these databases: PubMed, Embase, Web of Science, and Scopus. Keywords searched: sphingolipid and hypothalamus.	Non-English language.

**Table 2 ijms-22-05357-t002:** Characteristics and analysis of outcomes for in vitro studies.

			Outcomes Type			
Authors	Design	Treatment	Sphingolipid Extraction/Analysis Method	Sphingolipid Levels	Inflammatory Parameters	Neuropeptide Levels
Sortino et al., 1999	Hypothalamic GT1-7 neuronal cells.	TNF-α (20 ng/mL) for 15, 30 and 60 min. D609 (5 mg/mL) for 1 h.	Diacylglycerol kinase assay (commercially available kit)	↑ Cer-1-P (accumulation of ceramide) in GT1-7 cells stimulated with TNF-α. ↓ Cer-1-P in GT1-7 cells after pretreatment with D609.	↑ TNFR1 and TNFR2 in GT1-7 cells after TNF-α exposure	Uninformed
Davis et al., 2006	Wild-type mixed AH cultures (containing neurons and glia).	IL-1b (10–12 nM) for 30 s, 1, 2,5 and 10 min. C2-ceramide (5–10 lM) for 2,5,10 and 15 min.	Uninformed	↑ Src phosphorylation in the AH cultures after treatment with C2-Ceramide or IL-1β (concentration and time-dependent)	Uninformed	Uninformed
McFadden et al., 2014	PHN, N38HN, or R7HN cells.	Palmitate (PA) (5200 mM). C75 (570 mM). FSG67 (5160 mM). C89b (540 mM). TG (5300 nM). For 18 or 24 h.	Lipidomics	↑ Ceramide levels in PHD exposed to palmitate. ↓ Ceramide levels in PDH exposed to C75	↑ TNF-α, IL-1β, and IL-6 mRNA levels in PHN after exposed to C16:0 for 18 h. ↓ TNF-α, IL-1β mRNA levels in PHN exposed to C75 alone or in the presence of C16:0, FSG67, and C89b for 18 h.	Uninformed
Morselli et al., 2014a	N43 cells and BV2 cells.	Pretreated for the indicated time with 10^−8^ M E2 conjugated with fatty-acid-free BSA (albumin). 10^−8^ M E2 or 100 μM PA conjugated with BSA alone or in combination for 8 h. Cells were transfected with siRNAs targeting murine ERa or murine PGC-1a or with an unrelated control siRNA. N43 cells were infected with the FLAG-ERa (AdERa) and GFP adenoviral. A total of 4 h after virus exposure and 48 h later, cells were treated as indicated.	Liquid chromatography/electrospray ionization/tandem mass spectrometry	↑ Ceramide levels in N43 cells exposed to PA.	↑ TNF-α and IL-6 mRNA expression in N43 and BV2 cells exposed to PA for 8 h. ↓ TNF-α and IL-6 in N43 cells transfected with siRNA for PGC-1a, followed by treatment with PA. ↑ TNF-α and IL-6 in N43 cells transfected with siRNA for PGC-1a, followed by infection with AdGFP-Era followed by infection with AdGFP-ERa, with or without exposed to PA;	Uninformed
Morselli et al., 2014b	N43 hypothalamic cell line and primary neuronal cell cultures.	Palmitate. The steroid hormone 17-β estradiol (E2).	Mass spectrometry	Uninformed	↑ Inflammation in N43 cell line and primary neurons exposed to PA. ↓ Inflammation in N43 cell line and primary neurons were pretreatment with E2.	Uninformed
Silva et al., 2014	GT1-7 cells.	Leptin (40 µmol) for 12 h. Transfection with siRNA targeted to STAT3 or scrambled control siRNA.	Western blotting	↑ S1PR1 protein levels in GT1-7 cells treated with leptin, in a time-dependent manner. ↓ S1PR1 protein levels in GT1-7 cells transfected with STAT3 siRNA.	Uninformed	Uninformed
Campana et al.,2017	Hypothalamic GT1-7 neuronal cells.	Palmitate (1 mM) for 24 h. After, cells were stimulated with insulin (100 nM) for 5 min.	Liquid chromatography coupled with high-resolution mass spectrometry.	↑ Ceramide levels (C:16, C:18, C22, C24) in GT1-7 cells after treatment with palmitate. ↓ SPT2 blocked ceramide accumulation induced by palmitate in GT1-7 cells.	Uninformed	Uninformed
Dusaban et al., 2017	Astrocytes were isolated from P1-P3 postnatal WT and S1P3 KO mice.	siRNA for S1P1, S1P2, S1P3, and control siRNA. S1P2 antagonist JT-013 (1 μM). S1P treatment (0,5 μM, 6 h). S1p3 antagonist SPM-354 (5 μM, 15 min). FTY720 (100 nM, 6 h).	Quantitative-PCR	↑ S1P3 in WT astrocytes after scratch injury.	↑ COX2 and IL-6 levels in S1P3 KO astrocytes treated with S1P.	Uninformed
Tse, E. and Belsham, D., 2018	mHypoA-POMC/GFP-1, -2, -3, and -4 neurons.	10, 50, or 100 mM palmitate for 8 and 24 h. 50 mM methylpalmitate for 8 h. 10 nM insulin for 15 min. 1mM C16-ceramide for 8 h. 50 mM oleate for 8 h or cotreated with 50 μM palmitate and 50 μM oleate for 8 h. 100 mM myriocin or 50 mM L-cycloserine for 1 h. Inhibitors treatment for 1 h, [TAK-242, JNK (SP 600125), p38 MAP kinase (SB, 202190), or MEK1/2 (ERK1/2; PD 0325901) and PS- 1145 (10 mM)].	Uninformed	Uninformed	↑ mRNA levels of IL-6, IL-1B, TLR4, TNF-α, and NFkB after treatment with 50 mM palmitate for 8 h. ↓ IL6 mRNA expression after inhibition of TLR4 and pretreatment with PS 1145. ↑ IL-6 mRNA expression after pretreatment with SP 600125 and SB 202190. ↓ IL-6 mRNA expression after Oleate co-treatment. ↓ JNK phosphorylation in mHypoAPOMC/GFP-2 neurons pretreated with 50 mM palmitate and insulin stimulation.	↑ POMC mRNA expression in mHypoA-POMC/GFP-2 neurons with 50 mM palmitate for 8 h or 1 mM C16-ceramide for 8 h. ↓ POMC mRNA expression after pretreatments with 50mM SP 600125 (JNK inhibitor) or 10-mM PD 0325901 (ERK inhibitor). ↑ POMC mRNA expression in mHypoA-POMC/GFP-2 neurons with pretreatment with 100 mM myriocin or 50 mM L-cycloserine for 1 h. Cotreatment rescue PA induced POMC mRNA.
Sergi et al., 2018	mHypoE-N42 hypothalamic cell line. Primary hypothalamic culture from Sprague Dawley rats.	Palmitate (PA- 200 μM). Lauric acid (LA- 200 μM). Oleanolic acid (OA-200 or 125 μM). Eicosapentaenoic acid (EPA 200 or 125 μM). LPS 100 ng/mL. Synthesis, L-cycloserine (inhibitor of ceramide) 250 μM. TLR4 inhibitor (CLI-095) 1 μM. All for 6 h.	LC- ESI-MS/MS	↑ C16 ceramide after PA treatment. ↓ C16 ceramide when L-cycloserine was added in combination with PA. ↓ C16 ceramide in N42 neurons were treated with PA in the presence of OA or EPA.	↑ IL-6 and TNF-α expression in N42, N49 cells and hypothalamic cultures, after treatment with PA or LPS. ↓ IL-6 e TNF-α expression after treatment with L-cycloserine, in N42 cells compared with LPS group. ↓ IL-6 expression after treatment with LA in N42 cells. ↓ IL-6 and TNF-α expression after cotreatment with OA and EPA in N42 cells. ↓ IL-6 expression after treatment with PA combined with cycloserine compared to PA alone in N42 cells	Uninformed
Maldonado-Ruiz et al., 2019	Microglia primary culture from Wistar rats.	100 uM palmitic acid. 100 uM palmitoleic acid. 100 uM stearic acid. 100 uM linoleic acid. 25 uM N-hexanoyl-D-sphingosine (C6). 0.1 ug/mL LPS. All for 24 h.	Uninformed	Uninformed	↑ TNF-α, IL-6 and IL-1 after treatment with palmitate. ↑ TNF-α after treatment with stearic acid. ↑ IL-6 after treatment with palmitoleic acid. ↓ IL-6 after treatment with C6.	Uninformed

**Table 3 ijms-22-05357-t003:** Characteristics of in vivo studies.

			Outcome Type			
Authors	Design	Treatment	Weight gain	Food Intake	Adiposity (%)	Energy Expenditure
Gao et al., 2011	Male Sprague Dawley.	ICV injections of leptin (1 μg), cerulenin (40 μg), myriocin (4μg), N-hexanoyl-D-sphingosine (2.5 μg). CPT1-α adenovirus (overexpression or deletion).	Body weights after infusing leptin into the Arc overexpressing CPT1-α were significantly attenuated compared with the null condition. ↓ Body weight after infused myriocin or pretreatment with N-hexanoyl-D-sphingosine.	↑ Food intake in rats with overexpression of CPT1-α. ↓ Feeding after infusing leptin into the Arc overexpressing CPT1-α. ↑ Food intake in CPT1-α knockout mice after injections with leptin or cerulenin. ↓ Food intake after in infused myriocin or pretreatment with N-hexanoyl-D-sphingosine.	Uninformed	Uninformed
Borg et al., 2012	8 week old male mice of C57BL/6 and ob/ob lineage.	HFD for 12 weeks. Endurance exercise, once daily, 5 times a week for 6 weeks.	↑ Body mass in HFD mice Exercise training did not affect body mass in the HFD mice.	Uninformed	↑ Epididymal fat mass in HFD mice. Exercise training did not decrease epididymal fat mass.	Uninformed
Ramírez et al., 2013	Adult male mice of wild type and CPT1-α knockout.	Mice received an ICV of 5 mg or an IP of 10 mg of ghrelin. ICV administration of 4 mg of myriocin (1 h before ghrelin administration). ICV administration of 2.5 mg of C6:0 ceramide.	Uninformed	↑ Food intake after ghrelin IP injection in wild type mice. ↓ Food intake after ICV injection of myriocin (1 h before ghrelin administration). ↑ Food intake after ceramide injection in CPT1 KO mice.	Uninformed	Uninformed
Contreras et al., 2014	Male Sprague Dawley rats and lean and obese male Zucker rats (LZR, OZR).	ICV injections with C6 ceramide for 5 days. GRP78 adenoviral vectors for 6–9 days.	↑ Body weight after C6 injections for 5 days. ↓ Weight gain ceramide-induced after administration of GRP78 adenovirus. ↓ Body weight in OZR treated with GRP78 adenovirus.	Central treatment of C6 ceramide and GRP78 adenovirus did not affect feeding.	↑ Weight of gonadal and inguinal white adipose tissue promoted by central ceramide injections. ↓ Weight of gonadal and inguinal white adipose tissue promoted by GRP78 adenovirus. ↓ Weight of gonadal and inguinal white adipose tissue in OZR, promoted by GRP78 adenovirus.	Uninformed
Mera et al., 2014	Male Sprague Dawley.	Adeno-associated viral vector (AAV-CPT1AM—0.2 mL/min) injected into the VMH to increase CPT1AM.	↑ Body weight in CPT1AM rats after 20 days of AAV injection	↑ Food intake in CPT1AM rats after 20 days of AAV injection	↑ Adiposity in CPT1AM rats	Uninformed
Morselli et a., 2014a	8 week old male and female C57BL/6.	HFD 42% for 16 weeks.	↑ Body weight in males and females after consuming HFD for 16 weeks.	Uninformed	Uninformed	Uninformed
Morselli et al., 2014b	4 week old male and female C57BL/6.	HFD 42% for 4 weeks.	↑ Body weight of males and females after consuming HFD for 4 weeks.	Uninformed	Uninformed	Uninformed
Silva et al., 2014	5 week old male Wistar rats and 10 week old male Swiss, C57BL/6J, ob/ob and db/db mice.	HFD for 3 months. siRNA S1PR1/STAT3. ICV infusion: Leptin, SEW2871 (50 nM), S1P (50 ng), JSI124 (50 mM), FTY720 (50 mM).	↓ Body weight in obese rats with ICV injection of S1P. No difference in the total body weight was observed 48 h after S1PR1 siRNA injection.	↓ Food intake in obese and lean rats after injection of S1P or SEW2871. The JSI124 pretreatment, 30 min before S1P ICV injection, was sufficient to block the effect of S1P injection.	↓ Epididymal fat-pad weight in obese rats with ICV injection of S1P.	↑ Energy expenditure in rats with ICV injection of S1P. ↓ Energy expenditure in rats, after 4 h, with S1PR1 siRNA. Did not observe difference in the energy expenditure after S1P ICV injection in obese rats.
Campana et al., 2017	10 week old male Zucker rats (obese or lean) and Wistar.	ICV injections for 28 days with myriocin (300 nM) or vehicle in Zucker rats. In Wistar rats, either C2-ceramide or DH-C2-ceramide or vehicle (25nM) were acutely ICV injected. Prior to sacrifice, animals received an ICV injection of insulin (2 mUI) or saline.	Central myriocin treatment did not affect body weight in either obese or lean Zucker rats.	Central myriocin treatment did not affect food intake in either obese or lean Zucker rats.	Central myriocin treatment did not affect lean and fat body mass in either obese or lean Zucker rats.	Uninformed
Gao et al., 2017	6–10 week old male mice of LPL-knockout mice (GFAP-LPL^−/−^) and C57BL/6.	Standard chow diet or HFD for 10 weeks.	↑ Body weight in GFAP-LPL^−/−^ when fed an HFD.	↑ Food intake in GFAP-LPL^−/−^ when fed an HFD.	↑ Fat mass in GFAP-LPL^−/−^ when fed an HFD.	↓ Locomotor activity in LPL knockout with HFD. ↑ Energy expenditure in LPL knockout with HFD
Laperrousaz et al., 2017	8 week old male mice of MBHLpl, C57BL6/J, NexLpl−/−, Agrp−/−.	Adeno-associated viruses injected into the MBH to increase or deletion LPL.	↑ Body weight after LPL deletion. ↓ Body weight after LPL overexpression	There was no difference	↑ Adiposity in mice with LPL deletion.	↓ Locomotor activity and energy expenditure in mice with LPL deletion. ↑ RER in mice with LPL deletion.
Gonzalez-Garcıa et al., 2018	Female Sprague Dawley	Rats were bilaterally ovariectomized (OVX) or Sham-operated. ICV injections, after two weeks surgery: estradiol (1 nmol) for 6 days; myriocin (4 mg/day) or TUDCA (10 mg/day) for 6 days; GPR78 adenovirus; shSPTLC1 adenovirus. b3-AR specific antagonist SR59230A (3 mg/kg/day) was administrated subcutaneously twice a day, starting 2 days before the first ICV.	↑ Body weight in OVX rats after 15 days of surgery. ↓ Body weight in OVX rats after treatment with estradiol or myriocin or TUDCA or shSPTLC1 adenovirus or GPR78 adenovirus. ↑ Body weight in OVX rats after treatment with SR59230A.	↑ Food intake in OVX rats. ↓ Food intake in OVX rats after treatment with estradiol or myriocin. The other treatments did not affect dietary intake.	↑ Adiposity in OVX rats. ↓ Adiposity in OVX rats after treatment with estradiol or myriocin.	↑ Energy expenditure in OVX rats after treatment with estradiol. ↓ Respiratory quotient in OVX rats after treatment with estradiol.
Hernandez-Carretero et al., 2018	12 week old male mice of C57BL/6N.	LFD (low fat diet, 10% fat) for 18 weeks. HFD (high fat diet, 60% fat) for 18 weeks. SWD (switch diet from HFD to LFD). The SW group were fed HFD for 9 weeks and then switched to LFD for a further 9 weeks. Selective inhibitor of hematopoietic prostaglandin D synthase (HQL-79, 30mg/kg,) was administered by oral gavage for 5 days mice fed HFD.	↑ Body weight in HFD mice. ↓ Body weight in SWD mice. ↓ Body weight in HFD mice after treatment with HQL-79.	↓ Food intake in HFD fed mice treated with HQL-79	Uninformed	Uninformed
Maldonado-Ruiz et al., 2019	8 week old male and female Wistar rats.	Chow diet or cafeteria diet—Maternal Nutritional Programming Model (Male offspring from mothers exposed to Chow or CAF diets). Injected intradermically with 0.2 micrograms/kg of ghrelin or saline. For five days, ICV administration of: artificial cerebrospinal fluid (ACSF) (Control), 40 ug/uL palmitic acid (PAL), and 2 ug/uL lipopolysaccharide (LPS).	Uninformed	↑ Food intake in offspring programmed by the CAF diet. ↑ Food intake in spring programmed by maternal CAF diet after subcutaneous ghrelin injection. ↑ Food intake in offspring programmed by maternal CAF diet after LPS or PAL by ACSF injections.	Uninformed	Uninformed
Vozella et al., 2019	8 week old male mice of C57BL/6J.	Mice were fed a standard diet. Free feeding (FF). 12 h food deprivation (FD), 1 h refeeding after food deprivation (RF 1h). 6 h refeeding after food deprivation (RF 6h). On day 5, the mice were food-deprived for 12 h during the dark phase. The refeeding groups were food deprived for 12 h and then allowed to feed for 1 h or 6 h.	Uninformed	Uninformed	Uninformed	Uninformed
Heras et a., 2020	Female pups (small litters fed with HFD 45%).	Small litters (SL-4 pups/dam) and normal litter (NL- 12 pups/dam). HFD-45%; Myriocin or C6 treatment (2 ug/rat). Kisspeptin.	↑ Overnutrition female rats displayed increased body weight. C6 or Myriocin do not alter body weight. ↓ Underfed and Myriocin reduced body weight.	Not altered in Myriocin or C6 group	Uninformed	Uninformed
Lyu et al., 2020	*Pomc*-Cre mice (Stock No. 005965) and *ROSA*-tdTomato mice (Stock No. 007676).	Regular diet (RD) and high-fat diet (HFD- 60%) for 8 weeks. Unphosphorylated pRb (pRb∆P) lentivirus (cell cycle).	↑ Body weight in HFD-fed mice. ↓ Body weight in HFD-fed mice with telmisartan treatment. ↓ (pRb∆P) lentivirus body weight induced by HFD.	↑ Food intake in HFD-fed mice.	↑ Abdominal white fa in HFD-fed mice. ↓ Abdominal white fat in pRb∆P lentivirus rescue.	Uninformed
Rawish et al., 2020	6–8 week old male mice of C57BL/6N.	HFD (D12492) or normal fat diet. Once a day, for 6 or 13 weeks, mice received TELmisartan (8 mg/kg) or vehicle by oral gavage.	↑ Body weight in HFD-fed mice. ↓ Body weight in HFD-fed mice with telmisartan treatment.	↑ Food intake in HDF-fed mice. ↓ Food intake in HFD- Fed mice with telmisartan treatment.	↑ Fat mass in HFD-fed mice. ↓ Fat mass in HFD-fed mice with telmisartan treatment.	↓ Energy expenditure and locomotion in HDF-fed mice. ↑ Energy expenditure and locomotion in HDF-fed mice with telmisartan treatment, in dark periods.

**Table 4 ijms-22-05357-t004:** Analysis of outcomes for in vivo studies.

	Outcomes Type			
Authors	Sphingolipid Extraction/Analysis Method	Sphingolipid Levels	Inflammatory Parameters	Neuropeptide Levels
Gao et al., 2011	HPLC coupled with mass spectrometry	↑ Ceramide level in rats with overexpressing CPT-1a (under fasting condition). **↓** Ceramide level after leptin injection in rats with overexpression of CPT-1a. **↓** Ceramide level after myriocin injection in rats with overexpression of CPT-1a. **↓** Ceramide level in rats with CPT-1c deleted.	Uninformed	↑ NPY levels in rats with overexpression of CPT-1a. **↓** NPY levels after leptin injection in rats with overexpression of CPT-1a/ **↓** NPY levels after infused with myriocin in rats with overexpression of CPT-1a. The levels of AgRP and POMC were not altered.
Borg et al., 2012	Electrospray ionization-tandem mass spectrometry	↑ Ceramide species (18:0, 22:0 and 24:0) in the hypothalamus of HFD mice. ↑ Dihydroceramide and dihexosylceramide contents in the hypothalamus of HFD.	**↓** IκBα expression in HFD mice. ↑ IκBα with exercise training. Hypothalamic JNK signaling was not affected by obesity or exercise training	Uninformed
Ramírez et al., 2013	LC-ESI-MS/MS system	↑ Total ceramides and C:18 ceramides in the wild type mice after ghrelin injection.	ICV injection of myriocin (1 h before ghrelin administration) did not change the levels of inflammatory markers (TLR4, pIKKB, IKKB, NFKB)	↑ AgRP and NPY levels after ghrelin injection in wild-type mice. **↓** AgRP and NPY levels after pretreatment with myriocin (1 h before ghrelin administration). ↑ AgRP and NPY levels after ceramide injection in CPT1 KO mice.
Contreras et al., 2014	Liquid chromatography-electrospray ionization/multistage mass spectrometry system	↑ Concentration of C16 ceramide in the hypothalamus promoted by central ceramide injections. **↓** Concentration of ceramide C16 and C18 in the hypothalamus of OZR. Administration of the GRP78 adenovirus did not alter the levels of hypothalamic ceramide.	↑ IL-6, TNF-, and pIKKα/β expression in the hypothalamus after treatment with ceramide. Administration of the GRP78 adenovirus did not alter the expression of inflammatory markers in the hypothalamus.	Uninformed
Mera et al., 2014	Lipidomic analysis	Unaltered total ceramide levels. ↑ Ceramide species (C14:0 and C18:1), total lactosylceramides, total concentrations of sphingomyelin and dihydrosphingomyelin and lysophosphatidylcholine in CPT1AM animals. **↓** Total levels of lysophosphatidylethanolamine, plasmalogen-phosphatidylethanolamine, lysoplasmalogen, and lysophosphatidylserin in CPT1A animals.	Unaltered TNF-, IL-6, IL-1, and INOS mRNA expression. ↑ MCP1 mRNA expression in CPT1AM animals.	Unaltered POMC, CART, NPY, AgRP mRNA levels. ↑ NPY1R mRNA level in CPT1AM animals.
Morselli et a., 2014a	Liquid chromatography/electrospray ionization/tandem mass spectrometry	↑ Accumulation of ceramides and sphingomielin in the hypothalamus of male mice when compared to females, after HFD feeding. ↑ Glucosylceramide levels in males and females, after HFD feeding.	↑ TNF-α, IL-1β, and IL-6 levels in the hypothalamus of male, after HFD feeding. **↓** IL-10 in the hypothalamus of male, after HFD feeding	Uninformed
Morselli et al., 2014b	Mass spectrometry	↑ Ceramide, glucosylceramide, and sphingomyelin levels in male compared to female, consuming HFD for 4 weeks.	↑ TNF-α, IL-1β and IL-6 levels in male compared to female, after consuming HFD for 4 weeks.	Uninformed
Silva et al., 2014	Kit Assay, Western blot, RT-PCR	**↓** S1PR1 levels in the hypothalamus of obese rats.	↑ STAT3 phosphorylation in the hypothalamus of obese rats after injection of S1P.	↑ POMC mRNA in obese rats after injection of S1P or SEW2871, but not alter NPY.
Campana et al., 2017	Liquid chromatography/tandem mass spectrometry.	↑ Total ceramide levels in the hypothalamus of obese Zucker rats. **↓** Total ceramide levels in the hypothalamus of obese Zucker rats after ICV injection of myriocin.	Uninformed	Uninformed
Gao et al., 2017	Chromatographic separation and mass spectrometer	↑ Total ceramide levels and ceramide species (C18:0, C18:1 and C22) in the hypothalamus of GFAP-LPL^−/−^ mice.	↑ IBA1 in the hypothalamus of GFAP-LPL^−/−^ mice.	↑ AGRP in the GFAP-LPL^−/−^ mice. POMC cell number did not change
Laperrousaz et al., 2017	Liquid chromatography coupled with LC	**↓** Ceramide total levels and ceramide species (dC18/C16, dC18/C:18; DC18:0/C:20; dC18:00/C:22; dC18:0/C:24) in the hypothalamus with LPL deletion at 10 days after injection. **↓** Cers1 mRNA in the hypothalamus with LPL deletion at 10 days and 12 weeks after injection. ↑ Spt3, Cers2, Cers3 mRNA in the hypothalamus with LPL deletion at 12 weeks after injection.	Uninformed	NPY and AGRP did not affect
Gonzalez-Garcıa et al., 2018	Liquid chromatography/tandem mass spectrometry	↑ Ceramide levels in the hypothalamus after OVX. **↓** Ceramide levels in the hypothalamus of OVX rats after treatment with estradiol or myriocin.	Uninformed	Uninformed
Hernandez-Carretero et al., 2018	Electrospray ionization	↑ Ceramide species (C18:1) in adipose, muscle, and plasma of HDF mice. ↑ Ceramide species (C:20) in adipose, liver, and plasma of HDF mice. ↑ C18 dihydroceramide in adipose and plasma of HDF mice. ↑ C18:1 sphingosine in adipose and liver of HDF mice.	↑ F4/80 in HDF and SWD mice.	Uninformed
Maldonado-Ruiz et al., 2019	Uninformed	Uninformed	↑ IBA-1 in offspring programmed by a maternal CAF diet after ghrelin and palmitic acid injections. ↑ NFKB phosphorylation after LPS administration;	Uninformed
Vozella et al., 2019	Liquid chromatography/tandem mass spectrometry.	↓ Sphingosine, Sphingosine 1 Phosphate, Diidroxiceramide after 12 h of fasting. ↓ Hypothalamic levels of SA1P, S1P and SA after 12 h of fasting, which was partially (SA1P) or completely (S1P, SA) reversed after 6 h refeeding. ↓ Levels of dihydroceramide (d18:0/18:0) in the hypothalamus of food-deprived mice. ↓ S1pr1 transcription levels after 12 h of fasting an effect was rapidly and completely reversed by refeeding. ↓ Transcription levels of Sptlc2, Lass1, SphK2, Sphk1 after 12 h of fasting, which are partially (Sptlc2) or completely (Lass1, SphK2) reversed by refeeding.	Uninformed	Uninformed
Heras et a., 2020	High-performance liquid chromatography (HPLC)	↑ Hypothalamic total ceramide and ceramide species (CERC16, CERC18, CERC18:1, CERC:20, CERC(a)24:1, CERC14:0, CERC16:1, CERC20:1, CERC22:1, and CER24:2) in female rats subjected to early overnutrition.	Uninformed	Myriocin treatment did not change hypothalamic expression of Pomc, Cart, Npy, or Agrp. ↓ NPY levels after C6 treatment.
Lyu et al., 2020	RNAseq; qRT-PCR	↑Asah2, Cers2, and Elovl1 were in POMC neurons in the HFD group.	↑ Chemokines in POMC neurons in the HFD group.	Uninformed
Rawish et al., 2020	Liquid chromatography-mass	↑ Ceramide, cholesteryl ester, phosphatidylcholine, phosphatidylethanolamine, and sphingomyelin levels in plasma of HFD-fed mice. **↓** Ceramide levels in plasma of HFD-fed mice treated with telmisartan. ↑ Hypothalamic levels of ceramide (particularly Cer d36:1) (d18:1/18:0)) in HFD-fed mice. ↓ Hypothalamic levels of ceramide (particularly Cer d36:1) (d18:1/18:0)) in HFD-fed mice treated with telmisartan.	↑ TNF-α and Cxcl12 in HFD-fed mice. ↓ CXCL12 in HFD-fed mice after Telmisartan treatment. ↑ IL-4 and IL-6 levels in plasma of both HFD-fed mice and HFD-fed mice treated with telmisartan. ↑ IL-5 levels in plasma of HFD-fed mice with treatment of telmisartan.	Uninformed

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
