# Peer review of "The Role of Fatty Acids in Ceramide Pathways and Their Influence on Hypothalamic Regulation of Energy Balance: A Systematic Review"

_ijms, 2021, doi:10.3390/ijms22105357_

Round 1

Reviewer 1 Report

This review by Reginato and co-workers describes the role of hypothalamic ceramides and their metabolites on the control on energy balance. Particularly, Authors describe the deleterious role of ceramides in hypothalamic control of energy balance, proposing the modulation of sphingolipid metabolism as a promising avenue to tackle obesity. In general, the review reports on a really interesting a timely topic proving important insights on the role of ceramides in metabolic health. Nonetheless, in the reviewer’s opinion, the review presents several flaws. The manuscript is not particularly well and clearly written and requires some major changes to improve clarity and readability. Furthermore, and most importantly, the review lacks of spark and critical assessment of the data reported from other studies. Indeed, the authors only focus on describing the results without providing any details to explain these results and the underpinning mechanisms. Additionally, it would be important to highlight the role of different fatty acids in promoting ceramide accumulation, as fatty acid overload is not a sufficient explanation, particularly because not all fatty acids have the same impact of ceramide build-up within the hypothalamus. The mechanisms by which ceramides affect insulin signalling need to be described, for example, what is the role of protein kinase Cζ? Moreover, hypothalamic inflammation and ER stress are of particular importance in promoting hypothalamic insulin and leptin resistance. What is the role of ceramide in promoting inflammation and ER stress? Finally, hypothalamic fatty acid metabolism is pivotal in the regulation of ceramide levels, yet Authors do not make any reference to hypothalamic mitochondrial function which, in turn, is key in fatty acid catabolism. These are all key aspects that should be emphasised in the discussion of this manuscripts. Further details can be found in previous publications:  J. A.Chavez, S. A. Summers 2012, DOI:  10.1016/j.cmet.2012.04.002; D. Sergi, L. M. Williams 2020, DOI: 10.1093/nutrit/nuz056; A. Jais and J. C. Brüning, 2017, DOI: 10.1172/JCI88878; J. Cunarro et al., 2018, DOI: 10.3389/fendo.2018.00283.

Additional recommended revisions are reported below, but please bear in mind that these are only some of the example related to text readability and clarity that should be addressed by Authors.

Line 14: I would also say that this would enhance our understanding of the impact of diet of hypothalamic function.

Line 20: I would not say that ceramide leads to ectopic fat accumulation, instead ceramide is part of these lipid species that accumulates ectopically. Also, Authors should change “eating disorders” to “dysregulation of energy balance”.

Line 24: S1P should be defined.

Line 24-25: this sentence should be rephrased as it does not come across really clearly.

Line 35: please specify “type 2 diabetes mellitus”.

Line 38: “advance even more to prevent obesity” do authors mean: “improve our understanding of obesity pathophysiology to develop novel strategies to tackle obesity?”. The way the sentence is formulated at present is not really clear.

Line 44: please release “can be found” with “are located”.

Line 48: food intake can stand alone “consumption” is not required.

Line 50: “impaired” holds true for palmitic acid, however oleic acid and DHA exert a positive impact on insulin and leptin signalling.

Line 52: in the case of palmitic acid, it is important to mention inflammation and lipotoxicity.

Line 55: why are Authors referring to “saturated phospholipids”?

Line 61: please change “on” to “of”.

Line 86: this is particularly true for long-chain saturated fatty acids.

Line 104: “Inactivated” does not appear to be the correct term here.

Line 122: please change “physiology” to “pathophysiology”

Line 126: please change “under” to “in response to”.

Table 1. Can the resolution be increased?

Figure 4: same as table 1, the resolution should be increased.

Line 223 and 224: How can ceramide levels be assessed by qRT-PCR and Western blot? Same thing at line 230. These are not adequate techniques for this purpose.

I cannot find figure 2 in the text, was it uploaded separately or authors are referring to figure 2? Same thing for table 3 and 4.

Line 263: Please change “partially” to “partial”.

Line 271: Please change “blocked” with “inhibited”

Line 279: these in vitro studies, albeit proving potential mechanistic insights on the role of ceramide species in the regulation of energy balance, are not in vivo studies providing a physiological outcome such as food intake.

Line 336: it is also important to mention the ceramide is able to trigger hypothalamic ER stress. Contreras et al. (2014).

Citation number in the text do not correspond to reference order in the list of references. All the reference list should be corrected accordingly.

Line 349: it should read transcriptomic.

Line 361: “partial LPL” it looks like there is something missing. How does LPL affect ceramide levels?

Lines 366-369: this is a key example of what I found missing in the review. Here, the Authors do not provide any details on the role of ceramide in mediating the orexigenic effects of ghrelin. Authors should provide more details on the mechanisms underpinning the pathophysiological effects of ceramides throughout the text instead of merely reporting data from other studies. For example, in this specific case, these are specific questions to address: how does ghrelin regulate the levels of ceramides within the hypothalamus? Why is CPT1C important in this context? Please bare in mind that this suggestion holds true for the rest of the text, therefore I would recommend providing a more in-depth analysis of the results reported in this review. Authors may refer to the review reported above.

Line 395: ER stress should also be included in the list.

Line 401: Authors refer to humans, but to human data are reported. Thus, this needs to be changed.

Line 413: “on” should read “one”.

Line 421: please change “changes” to “regulation”.

Lines 421-426: Instead I would say that the best way to validate gene and protein expression data, would be via a direct assessment of ceramide levels using analytical techniques such as LC-MS. As I have already suggested, qPCR and Western blot cannot be used to assess ceramide levels.

Throughout the text it is of particular importance to specify whether Authors are referring to CPT1c or other isoforms because the function of these CPT1 isoforms may differ.

Author Response

We appreciate all comments from Reviewer 1. We have included the responses in a pdf file, together with the responses to the comments of the Reviewer 2. Please see the attachment.

Reviewer 2 Report

This submission is a review article that highlights the role that sphingolipids play in hypothalamic metabolism control, and makes the case for the exploration of sphingolipid metabolic manipulation in obesity treatment.  The review is well written and supports the conclusions drawn by the authors.  There are however some slight clarification points that could be made to produce a more complete and accurate review.  Overall, the small omissions do not reach the level that they compromise the work completely, and with moderate revision this article deserves publication.

Minor points:

English language issues:  there are minor issues with plural usage/ grammar in the text- for example line 90-91 "excessive ceramides levels" should read excessive ceramide levels.  Additionally line 78 should read known as the "salvage pathway".  line 121 "sphingolipids metabolism" should read sphingolipid metabolism.  Minor errors like this are sprinkled through the review.

Nomenclature:  Sphingosine-1-phosphate is the correct nomenclature, not sphingosine 1-phosphate.

Incomplete or missing context:

lines 77-84 suggest that the salvage pathway is an inducible event alone, however every endocytic event is accompanied by degradation of complex sphingolipids to their base building block sphingosine inside the endocytic compartment.  This resulting sphingosine reaches the ER where is is either metabolized buy SGPL or utilized in the salvage pathway.  This is a fundamental cellular process that occurs at all times in the cells and does not require any signaling event such as this provided by the authors.  Suggest clarification.

lines 89-96 suggest that the chain length (which is not fully explained), regulated by CarS isoform expression and specificity, can be associated with negative biological events.  To support this conclusion (which is supported by the literature, just not the literature used by the authors) the authors point to beneficial effects of SPT inhibition (not chain length or CerS isoform specific) and global ceramide changes associated with lipotoxicity and immune dysfunction (again not related to chain length and CarS isoform).

Line 103- S1P is generated from sphingosine by sphingosine kinases.  Ceramide has no inherent catalytic activity, and thus cannot "generate" anything.  Additionally, SGPL (or S1PL in authors terminology does not inactivate S1P, it irreversibly breaks the molecule down.  This section needs reworked to be more accurate.

Line 198-201:  This is confusing.  The authors state" The main interest of this systematic review was the evaluation of ceramide/sphingo-198 sine 1-phosphate originated from de novo ceramide or salvage pathway (Figure 3) and their influence on hypothalamic energy balance control, especially under fatty acids interventions."  The use of "or" suggests it is one or the other, yet no final conclusion is presented to support this distinction.

Author Response

We appreciate all comments from Reviewer 2. We have included the responses in a pdf file, together with the responses to the comments of the Reviewer 1. Please see the attachment.

Round 2

Reviewer 1 Report

Thank you for addressing my comments. Please reconsider the conclusion as I noticed some high degree of similarity with regard to "the consumption of saturated fatty acids in combination with refined carbohydrates and sugar" which was reported in previously published work. Thus, either authors make some amendments to this sentence or should cite the original source. In line 650  I am not sure what authors mean by "GPR78 treatment", in the manuscript authors are referring to, the overexpression of the chaperone GPR78 improved the metabolic phenotype of the rodents.

Line 631: the sentence is incomplete.

Line 622: this is also important in mediating the metabolic effects of insulin.

The fact that qPCR and Wester blot are not suitable methods for ceramide quantification should also be mentioned in the the rest of the text, not only in the discussion.

Round 3

Reviewer 1 Report

"GRP78 adenovirus" does not explain the concept Authors would like to get across. It would be sufficient to refer to GPR78 overexpression. "... which is reversed by GRP78 overexpression..."
